# High-resolution structure determination of sub-100 kDa complexes using conventional cryo-EM

Mark A. Herzik Jr. [1,2], Mengyu Wu[1] & Gabriel C. Lander [1]

Determining high-resolution structures of biological macromolecules amassing less than 100 kilodaltons (kDa) has been a longstanding goal of the cryo-electron microscopy (cryo-EM) community. While the Volta phase plate has enabled visualization of specimens in this size range, this instrumentation is not yet fully automated and can present technical challenges. Here, we show that conventional defocus-based cryo-EM methodologies can be used to determine high-resolution structures of specimens amassing less than 100 kDa using a transmission electron microscope operating at 200 keV coupled with a direct electron detector. Our ~2.7 Å structure of alcohol dehydrogenase (82 kDa) proves that bound ligands can be resolved with high fidelity to enable investigation of drug-target interactions. Our ~2.8 Å and ~3.2 Å structures of methemoglobin demonstrate that distinct conformational states can be identified within a dataset for proteins as small as 64 kDa. Furthermore, we provide the sub-nanometer cryo-EM structure of a sub-50 kDa protein.

[1] Department of Integrative Structural and Computational Biology, The Scripps Research Institute, La Jolla, CA 92037, USA. [2] Present address: Department of Chemistry and Biochemistry, University of California, San Diego, La Jolla, CA 92093, USA. These authors contributed equally: Mark A. Herzik, Jr. and Mengyu Wu.  Correspondence and requests for materials should be addressed to G.C.L. (email: glander@scripps.edu)

In recent years, technical advances in cryo-electron microscopy (cryo-EM) single-particle analysis (SPA) have propelled the technique towards the forefront of structural biology, enabling the direct visualization of biological macromolecules in near-native states at increasingly higher resolutions[1]. Notably, cryo-EM enables three-dimensional (3D) structure determination of biological specimens in a vitrified state without the requirement of crystallization[2], which has not only greatly increased the throughput of high-resolution structure determination, but has also allowed for the 3D visualization of macromolecular complexes previously deemed intractable for structural studies due to size, conformational heterogeneity, and/or compositional variability[3–5]. Indeed, determining ~3 Å resolution reconstructions of stable, conformationally and/or compositionally homogeneous specimens by SPA has become almost routine, with an increasing number of structures at ~2 Å resolution or better now being reported[6–8]. This resolution regime has also expanded the potential of cryo-EM SPA for structure-based drug design, particularly for targets that are less amenable to other structure determination techniques due to limited sample quantity or recalcitrance to crystallization.

Despite these advances, the radiation sensitivity of ice-embedded proteins and the low signal-to-noise ratio (SNR) of cryo-EM images nonetheless impede routine structure determination for all samples, and specimen size remains a considerable limiting factor in cryo-EM SPA. Indeed, although SPA reconstructions of molecules as small as 38 kilodaltons (kDa) have been theorized to be achievable[9], this feat has yet to be realized. To date, only three macromolecular complexes smaller than 100 kDa have been resolved to high resolution (i.e., better than 4 Å) using cryo-EM SPA: the ~3.8 Å resolution reconstruction of 93 kDa isocitrate dehydrogenase[10], the ~3.2 Å resolution reconstruction of 64 kDa methemoglobin (mHb)[5], and the ~3.2 Å resolution reconstruction of 52 kDa streptavidin[11]. Due to the limited success in imaging smaller macromolecules by cryo-EM, the technique has primarily been used to visualize large complexes, with ~99% of all cryo-EM SPA reconstructions resolved to better than 5 Å resolution comprising macromolecules amassing >200 kDa.

We previously demonstrated that a transmission electron microscope (TEM) operating at 200 keV equipped with a K2 Summit direct electron detector (DED) could be used to resolve a ~150 kDa protein complex to ~2.6 Å using conventional defocus-based SPA methods[12]. Here, we expand upon our previous results and show that biological specimens amassing <100 kDa can be resolved to better than 3 Å resolution using similar imaging approaches. The resulting reconstructions possess well-resolved density for bound cofactors, metal ligands, as well as ordered water molecules. We further demonstrate that conformational heterogeneity in specimens of this size range can be discerned. Finally, we provide the sub-nanometer single-particle cryo-EM structure of a sub-50 kDa macromolecular complex – the 43 kDa catalytic domain of protein kinase A.

## Results

**Conventional defocus-based single-particle cryo-EM.** Our prior success in resolving the structure of a sub-200 kDa complex to better than 3 Å resolution demonstrated that conventional defocus-based methodologies provided sufficient SNR to confidently assign 3D orientations to biological specimens that were previously thought to be too small to image[12]. This prompted us to investigate whether high-resolution reconstructions of macromolecules <100 kDa in size could be achieved with conventional imaging approaches, provided sufficiently thin ice and high particle density could be attained to maximize the accuracy of contrast transfer function estimation. All specimens in this study were imaged using a base model (i.e., excluding imaging accessories such as a phase plate or energy filter) Talos Arctica TEM equipped with a K2 Summit DED operating in counting mode using the Leginon[13] automated data collection software. TEM column alignments were performed as described previously[12,14] with the following modification to maximize parallel illumination: after minimizing the spread of gold powder diffraction rings, the camera length was increased to 5.7 m and both the size of the caustic spot and the diffraction astigmatism were iteratively minimized in order to optimize parallel illumination of the sample (see Methods).

**2.7 Å resolution structure of 82 kDa alcohol dehydrogenase.** We first decided to target the 82 kDa homodimeric enzyme alcohol dehydrogenase (ADH)[15], as the structure of this complex had been previously determined to near-atomic resolution (e.g., better than 1.3 Å resolution) by X-ray diffraction[16], providing an atomic model for validation of our results. Notably, as ADH is a nicotinamide adenine dinucleotide (NADH)-binding protein, part of our motivation for selecting this enzyme for structure determination was to test our ability to resolve the bound NADH cofactor, as this would serve as important proof-of-principle for the utility of SPA for structure-based drug design of similarly sized samples.

We further purified commercially sourced horse liver ADH (Sigma Aldrich) to homogeneity and prepared grids of the frozen-hydrated specimen for cryo-EM data collection (see Methods) (Supplementary Fig. 1). Despite the small size of ADH, multiple views of the homodimer could be clearly distinguished in the aligned images even when using a nominal defocus range of −0.5 μm to −1.6 μm, contrary to previous estimations that significant underfocus would be required to image small (<200 kDa) macromolecules using conventional EM[17] (Supplementary Fig. 2). However, the large disparity in orthogonal dimensions of ADH (~100 Å versus ~40 Å, see Fig. 1) posed a challenge during two-dimensional (2D) classification. To overcome this, two successive rounds of RELION[18,19] reference-free 2D classification were performed, the first using a 110 Å soft circular mask to identify particles comprising the longer side/tilted views, followed by a second round of 2D classification of the remaining data using a 60 Å soft mask to identify particles comprising the end-on views (Fig. 1a, Supplementary Fig. 2e). These combined particles were subsequently subjected to several rounds of 3D auto-refinement and classification with C2 symmetry enforced to yield a final ~2.7 Å reconstruction, as determined by gold-standard Fourier shell correlation (FSC) at 0.143[20–22] (see Methods) (Fig. 1, Supplementary Table 1, and Supplementary Fig. 2). These same particles refined to ~3.1 Å resolution without imposing symmetry (Supplementary Fig. 2).

Inspection of the map revealed side-chain density for most of the ADH polypeptide with pronounced backbone features for the entire macromolecule. Indeed, local resolution estimates[23] indicated that the majority of the map was resolved to better than ~3.3 Å resolution, with the core of ADH resolved to ~2.6 Å resolution (Fig. 1b). Moreover, there is unambiguous density for the bound NADH cofactor, a catalytic zinc ion within the active site of each protomer, as well as a structural zinc ion coordinated by four cysteine residues (Fig. 1). Importantly, the positions of these ligands are corroborated by the X-ray crystal structure of horse ADH (PDB ID: 2JHF [https://doi.org/10.2210/pdb2JHF/pdb]). This reconstruction clearly establishes that <100 kDa complexes can be resolved to better than 3 Å resolution using conventional approaches using a 200 keV TEM without the need for a phase plate or energy filter. Furthermore, our ability to confidently identify bound ligands and cofactors within our ADH

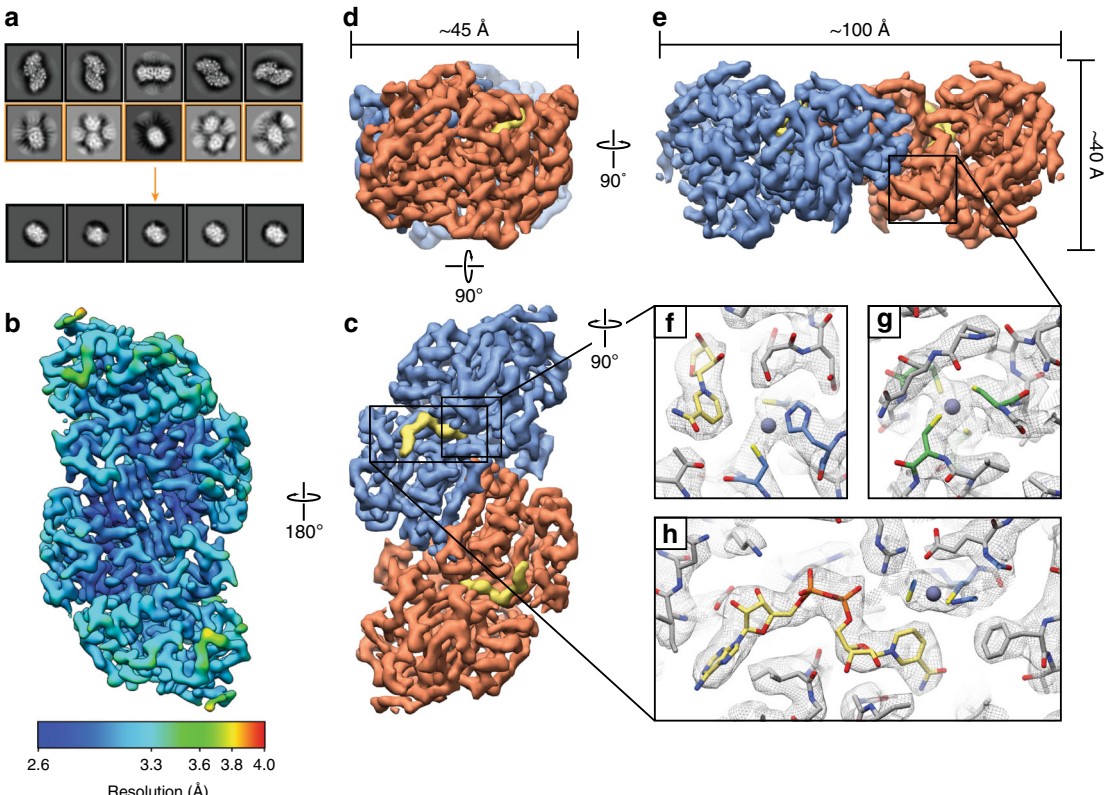

**Fig. 1** 2.7 Å resolution cryo-EM reconstruction of 82 kDa horse liver alcohol dehydrogenase. **a** Representative reference-free 2D class averages of horse liver alcohol dehydrogenase (ADH). Particles comprising the 2D classes highlighted in yellow were subsequently further classified using a smaller soft circular mask (see methods). **b** Final cryo-EM reconstruction colored by estimated local resolution estimated with BSOFT[23]. **c-e** Orthogonal views of the ADH EM density (colored by subunit) showing the disparity in particle dimensions. The segmented NADH EM density is shown in yellow. **f-h** Zoomed-in views of the EM density (gray mesh) for the ADH active-site zinc, structural zinc site, and active site, respectively. Residues involved in coordinating the active-site zinc (blue), the structural site zinc (green) or interacting with NADH (yellow) are shown in stick representation. The zinc atoms are shown as purple spheres

EM density effectively demonstrates the utility of conventional SPA methods for structure-based drug design and other small-molecule research involving similarly sized specimens.

**Two states of methemoglobin at ~2.8 Å and ~3.2 Å resolution.** Khoshouei et al.[5] recently demonstrated that a TEM operating at 300 keV combined with a Volta phase plate (VPP) and quantum energy filter could be used to resolve human hemoglobin (Hb), a ~64-kDa heterotetrameric heme-containing protein, to ~3.2 Å resolution. It has been speculated that the increased SNR afforded by the VPP was integral for structure determination of Hb[5]. However, our ability to resolve ADH to high resolution indicated that conventional defocus-based imaging methodologies provided sufficient SNR for structure determination, and that we had not yet reached the size limit of this approach. We therefore sought to determine the structure of Hb using similar strategies. Briefly, vitrified Hb specimens were prepared from lyophilized human Hb (Sigma Aldrich) (see Methods). UV-VIS absorbance measurements of the solubilized Hb sample confirmed that the bound heme was indeed in the ferric ($Fe^{3+}$) oxidation state, hereby referred to as methemoglobin (metHb).

Images of frozen-hydrated metHb were collected similarly as described for ADH (see Methods) (Supplementary Fig. 3). Notably, orthogonal views of metHb were discernible by eye (Fig. 2b), even in micrographs imaged using underfocus values as low as ~700 nm. RELION reference-free 2D classification yielded detailed class averages exhibiting secondary-structural elements and various recognizable views of tetrameric metHb (Fig. 2c).

Three parallel 3D classifications were performed to select for unique particles comprising the best-resolved classes across each classification (see Methods, Supplementary Fig. 3). An additional round of no-alignment 3D classification revealed two distinct conformational states of metHb: state 1 (~2.8 Å resolution) closely matches the "Near R2" state previously described by Shibayama[24] et al. 2014 (Cαroot-mean-square-deviation (RMSD) 0.4 Å, PDB ID: 4N7P [https://doi.org/10.2210/pdb4N7P/pdb]) (Fig. 3a) and state 2 (~3.2 Å resolution) agrees well with the "Between R and R2" state (CαRMSD 0.5 Å, PDB ID: 4N7N [https://doi.org/10.2210/pdb4N7N/pdb]) (Fig. 3b). Comparison of the two states following superposition of an αβ dimer from each molecule revealed an ~7° rigid-body rotation of one αβ dimer with respect to the other with the rotation axis centered about the dimer–dimer interface, similar to those movements previously described[24] (Fig. 3c). These results demonstrate that distinct, biologically relevant conformational states of a sub-100 kDa complex can feasibly be resolved to high-resolution using single-particle cryo-EM.

Local resolution estimates indicated that most of both maps were resolved to better than ~3.5 Å, which is consistent with our ability to resolve side-chain densities throughout the maps (Fig. 2f). Notably, the heme pockets are estimated to be the best-resolved regions in both states (Supplementary Fig. 3), and the quality of the heme densities in both α and β subunits of state 1 enabled us to confidently discern the location of the vinyl groups extending from the porphyrin ring to unambiguously assign the orientation of the heme moiety in each pocket

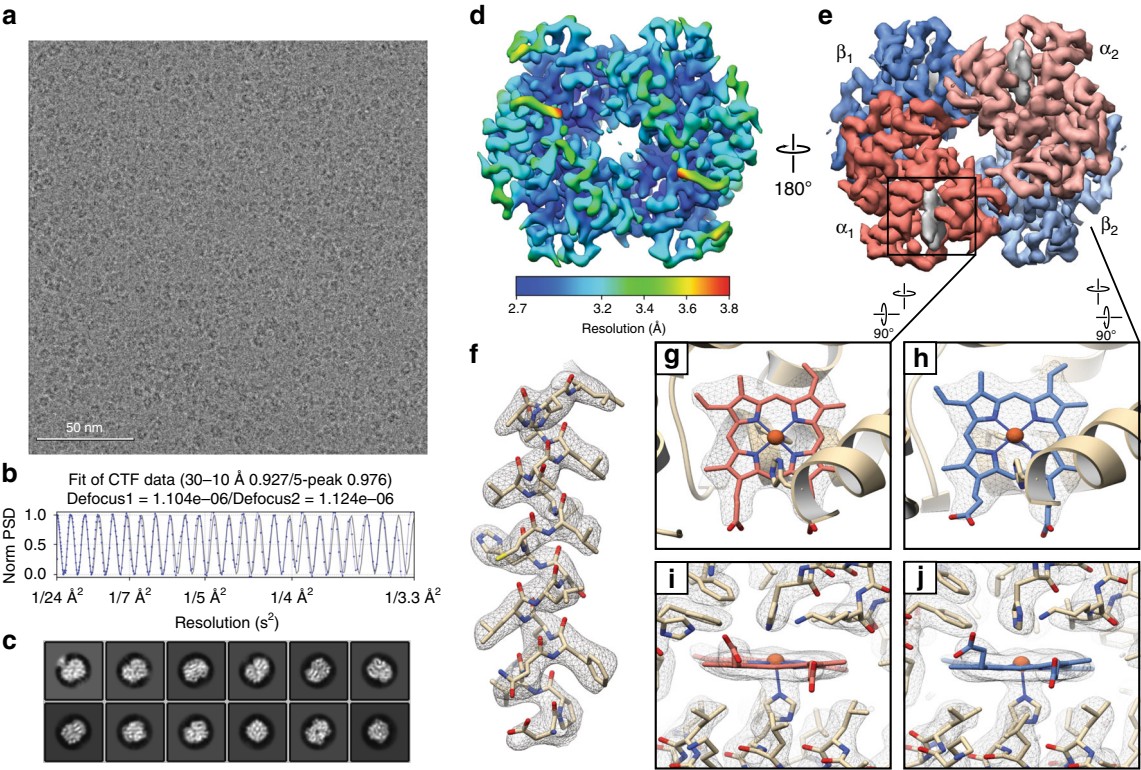

**Fig. 2** 2.8 Å resolution cryo-EM reconstruction of ~64 kDa human methemoglobin. **a** Representative motion-corrected electron micrograph of human methemoglobin (metHb) embedded in vitreous ice recorded at ~1.1 μm defocus (scale bar, 50 nm). **b** 1-dimensional plot of the contrast transfer function (CTF) Thon rings (black line) and the CTF estimated with CTFFIND4[33] (blue line). **c** Representative reference-free 2D class averages showing secondary structure elements. **d, e** Final ~2.8 Å resolution metHb cryo-EM density colored by local resolution (estimated using BSOFT[23]) and subunit with the segmented EM density for the heme cofactors colored gray, respectively. **f** EM density (gray mesh) zoned 2 Å around an α-helix comprising residues 94–113 from the α subunit. **g–j** EM density in the vicinity of the heme cofactors from subunit α1 and β2. Lower panels are 90° rotations of upper panels. Residues are shown in stick representation (colored wheat) and the heme cofactors are colored according to the subunit coloring in **e**. The heme iron atoms are shown as orange spheres

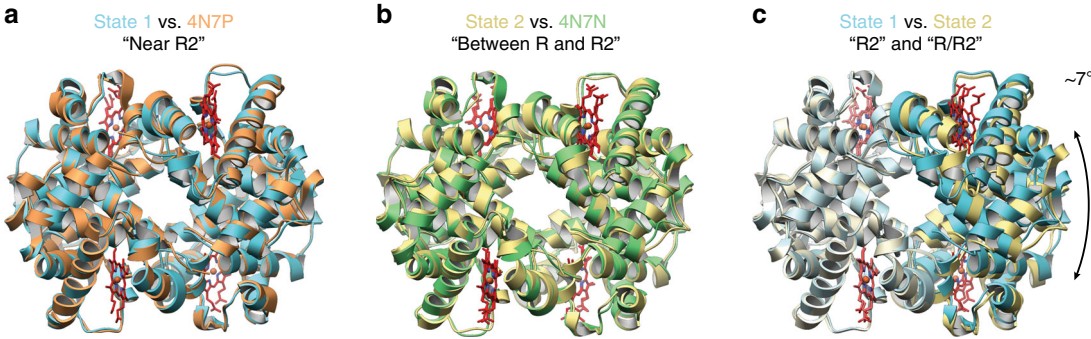

**Fig. 3** Multiple conformational states of metHb determined using single-particle cryo-EM. **a** Cartoon representation of metHb state 1 (~2.8 Å resolution, blue) superposed with PDB ID: 4N7P [10.2210/pdb4N7P/pdb] (chains A–D, orange). **b** Cartoon representation of metHb state 2 (~3.2 Å resolution, yellow) superposed with PDB ID: 4N7N [10.2210/pdb4N7N/pdb] (chains E-H, green). **c** Superposing the α1 and β1 subunits of states 1 and 2 emphasizes the difference between these two conformations as a ~7° rotation of the α2 and β2 subunits

(Fig. 2g, h). Furthermore, the well-resolved regions of state 1 contain putative density for ordered water molecules that are conserved between those observed in a previously obtained crystal structure (PDB ID: 2DN1 [https://doi.org/10.2210/pdb2DN1/pdb]). Our ability to discern subtle conformational differences in a target as small as 64 kDa effectively underscores the utility of this approach in examining the conformational dynamics of similarly small biological systems.

**Towards a high-resolution structure of a 43 kDa complex.** Processing of the metHb dataset required two successive rounds of 2D classification, partially due to the presence of contaminating ~32-kDa αβ heterodimers, which surprisingly accounted for ~20% of the data (Supplementary Fig. 3). Subsequent 2D classification of these particles yielded class averages with defined secondary-structural elements (Fig. 4). Furthermore, 2D projections of a simulated EM density generated using an αβ

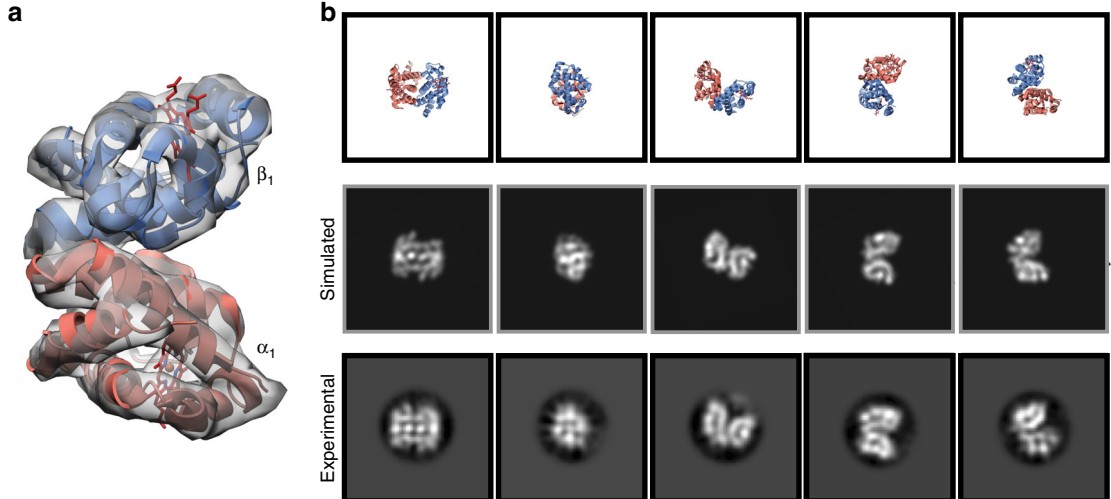

**Fig. 4** 2D classification yields classes of ~32 kDa hemoglobin αβ heterodimer. **a** Simulated EM density of the hemoglobin αβ heterodimer low-pass filtered to 8 Å resolution shown as a gray transparent surface. The αβ heterodimer atomic model is shown as ribbon cartoon and the hemes are shown as red sticks. **b** Views of the hemoglobin αβ heterodimer atomic model (top) and 2D projections of the simulated hemoglobin αβ heterodimer EM density (middle) corresponding to the class averages obtained from 2D classification (bottom) (see Fig. 2)

dimer from PDB ID: 4N7N [https://doi.org/10.2210/pdb4N7N/pdb] exhibited striking similarity to these class averages (Fig. 4). However, generation of a 3D reconstruction was unsuccessful, likely due to preferred orientation and/or poor SNR of the particles in vitreous ice. Despite this, our results for the metHb tetramer indicated that we had not yet reached the size limit resolvable by our imaging methodologies, and the promising 2D averages of the metHb dimers prompted us to continue investigating the lower molecular weight limit of cryo-EM SPA.

We next decided to pursue the structure of the asymmetric ~43 kDa catalytic domain of protein kinase A (PKA$_c$) bound to ATP, manganese, and IP20, an inhibitory pseudo-peptide substrate that has been previously shown to stabilize PKA$_c$[25]. EM grid preparation and imaging of frozen-hydrated inhibited PKA$_c$ (iPKA$_c$) were performed similarly as described for ADH and metHb with minor modifications (see Methods). Remarkably, given the small size and dimensions of iPKA$_c$ (65 × 40 × 40 Å), particles were clearly discernible in the aligned micrographs even at modest underfocus values (e.g., <1000 nm) (Fig. 5).

Reference-free 2D classification of gaussian-picked particles yielded featureful 2D class averages representing numerous views of iPKA$_c$ wherein the N- and C-terminal lobes of the complex, as well as secondary-structural elements, could be clearly discerned. Consistent with these observations, ab initio 3D model generation using cryoSPARC[26] yielded a volume resembling low-resolution iPKA$_c$ (Supplementary Fig. 4c). However, subsequent 3D auto-refinement yielded a >4 Å resolution reconstruction of iPKA$_c$ exhibiting pronounced features consistent with preferred orientation of particles at the air-water interface (e.g., stretching along the axis orthogonal to the dominant view) and over-fitting (e.g., streaking density extending beyond the masked region), indicating that the FSC-reported resolution was substantially inflated. Examination of the Euler distribution plot confirmed that regions of Euler space were unaccounted for (Supplementary Fig. 4c). In an effort to obtain these missing views we imaged the specimen at a tilt angle of 30° using methodologies previously described[27] (see Methods). 2D classification of particles extracted from the tilted data gave rise to averages comprising views that were not observed in the untilted data (Fig. 5c). However, the overall visual quality of the class averages from the tilted data were lower than those of the untilted data, presumably due to the combination of

increased beam-induced motion and decreased SNR of the particles resulting from imaging a tilted specimen[27]. Nonetheless, our ability to obtain featureful iPKA$_c$ 2D class averages from tilted micrographs indicates that this imaging scheme can be applied to biological macromolecules of a wide range of sizes.

3D auto-refinement of the combined particles from the tilted and untilted data yielded a ~6 Å resolution reconstruction of sufficient quality to discern tertiary structure elements consistent with the iPKA$_c$ kinase fold. Although this reconstruction appeared to be more isotropic in resolution than the untilted data, the secondary structure elements of the EM density are smooth and featureless. Numerous computational efforts were employed (see Methods) to improve particle alignment and obtain a high-resolution reconstruction of iPKA$_c$, but we were unable to overcome the resolution-limiting, gross misalignment of the particles (Supplementary Fig. 5). Given these complications arising from low image SNR, our work with iPKA$_c$ suggests that the use of a VPP and/or energy filter may potentially benefit imaging of targets of comparably small sizes (i.e., <50 kDa) by boosting observable contrast while preserving high spatial frequency information. However, the untilted 2D class averages demonstrate that detailed projections of <50 kDa particles are resolvable by conventional cryo-EM SPA, and that algorithmic development may enable a high-resolution structure to be produced from these data.

## Discussion

The VPP has previously demonstrated excellent utility in resolving the structures of some small biological specimens[5,11], and consequently is now widely perceived as a necessity for resolving smaller targets. The results presented in this study define a new frontier for target sizes that can feasibly be resolved using cryo-EM without the need for a phase plate. Although the use of significant underfocus (e.g., >2000 nm) has been speculated to be required for imaging macromolecules <200 kDa by conventional cryo-EM SPA, it is clear from this work that even molecules as small as ~43 kDa can be resolved with modest underfocus (e.g., <1000 nm) provided the vitreous ice encompassing the molecule of interest is sufficiently thin (see Supplementary Note 1, Supplementary Fig. 6). Moreover, the results of our work with ADH and mHb demonstrate that small-molecule ligands and discrete

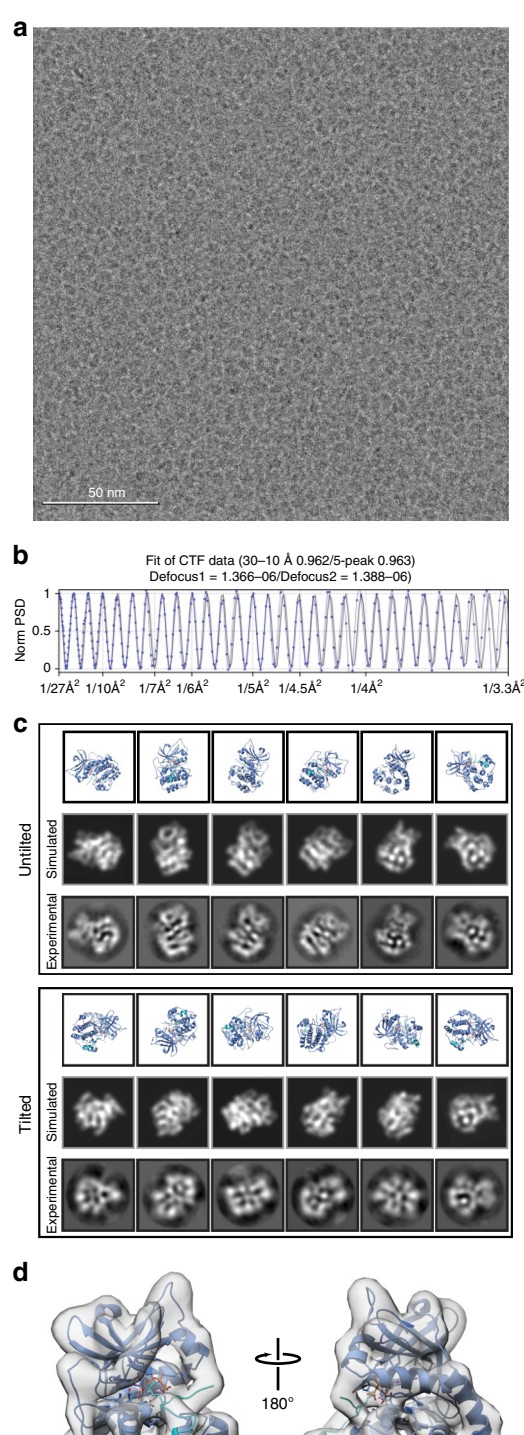

**Fig. 5** Towards a high-resolution cryo-EM reconstruction of the ~43 kDa isolable kinase domain of protein kinase A. **a** Representative motion-corrected electron micrograph of the catalytic subunit of protein kinase A bound to IP20 (iPKA$_c$) embedded in vitreous ice recorded at ~1.3 μm defocus (scale bar, 50 nm). **b** 1-dimensional plot of the contrast transfer function (CTF) Thon rings (black line) and the CTF estimated with CTFFIND4[33] (blue line). **c** Views of the iPKA$_c$ atomic model (top, PDB ID: 1ATP [10.2210/pdb1ATP/pdb], blue cartoon) and 2D projections of the simulated iPKA$_c$ EM density (middle) corresponding to the class averages obtained from 2D classification (bottom). **d** Final ~6 Å iPKA$_c$ EM density shown as a transparent gray surface with the fitted atomic model (PDB ID: 1ATP [10.2210/pdb1ATP/pdb]) shown as a blue cartoon. ATP is shown as sticks and IP20 is colored light blue

arising from the inherently limited SNR of the molecule in vitreous ice that was further confounded by molecule asymmetry, manifesting as gross misalignment within the 3D reconstruction. It is likely that <100 kDa membrane proteins solubilized in detergents or nanodiscs will similarly suffer from limited SNR-induced misalignment due to the presence of the disordered molecules that surround the transmembrane domains. Future developments in detector technology and/or image-processing algorithms will be integral for high-resolution structure determination of these small protein complexes that approach the theoretical size limit of the technology. These improvements will also be of great relevance to structural studies of small membrane-embedded proteins, the SNRs for which are detrimentally impacted by the presence of disordered detergent micelles or nanodiscs. Our work further propels an important trajectory for cryo-EM SPA and indicates that high-resolution structure determination of complexes approaching, or even exceeding, the theoretical size limit of cryo-EM SPA will likely be realized in the near future.

## Methods

**Sample handling and protein purification**. Lyophilized horse liver alcohol dehydrogenase (Sigma Aldrich) (ADH) was solubilized in 20 mM HEPES (pH 7.5), 10 mM NaCl, 1 mM TCEP (Buffer A), and dialyzed overnight against the same buffer at 4 °C. Dialyzed protein was then diluted two-fold with Buffer A lacking NaCl and immediately loaded onto a HiTrap SP HP (GE Life Sciences) that had been equilibrated in Buffer A. ADH was eluted using a gradient from 0 to 100% Buffer B (20 mM HEPES (pH 7.5), 500 mM NaCl, 1 mM TCEP) over 15 mL while collecting 500 μL fractions. Fractions containing ADH were pooled and dialyzed overnight at 4 °C against 20 mM Tris (pH 8.5), 100 mM NaCl, 1 mM TCEP, and 0.5 mM NADH (Buffer C). ADH was then concentrated to ~300 μL using a 30,000 MWCO spin concentrator and subjected to size-exclusion chromatography using a Superdex 10/300 GL (GE Life Sciences) that had been equilibrated in Buffer C. Pure ADH was pooled and concentrated to ~2.5 mg mL$^{-1}$ and used immediately for cryo-EM grid preparation.

Lyophilized human hemoglobin (Sigma Aldrich) (Hb) was solubilized in PBS (pH 7.5) to a final concentration of ~12 mg mL$^{-1}$ and used without further modifications. A UV-VIS absorbance spectrum of the solubilized protein indicated a Soret maximum at 406 nm, consistent with previous measurements of methemoglobin (metHb), but with two small peaks in the α/β region at 576 nm and 541 nm, respectively, potentially indicating a small amount of deoxyhemoglobin in the sample.

The catalytic subunit of protein kinase A (PKA) bound to ATP, manganese, and inhibitor peptide 20[28] was donated by P. Aoto of the Taylor lab (UCSD) and used without modification.

**Cryo-EM grid preparation**. A volume of 3 μL of purified ADH (~2.5 mg mL$^{-1}$), metHb (~12 mg mL$^{-1}$), or inhibited PKA (~5 mg mL$^{-1}$) were dispensed on UltrAuFoil R1.2/1.3 300-mesh grids (Electron Microscopy Services) that had been freshly plasma cleaned using a Solarus plasma cleaner (Gatan, Inc.) with a 75% argon/25% oxygen atmosphere at 15 Watts for 6 s. Grids were blotted manually[29] using a custom-built manual plunger in a cold room (≥95% relative humidity, 4 °C)[12]. Samples were blotted for 4–5 s with Whatman No. 1 filter paper immediately before plunge freezing in liquid ethane cooled by liquid nitrogen. We aimed to achieve a particle concentration that maximized the number of particles contained within the holes without resulting in overlapping particles and/or

conformational states within a single sample, respectively, can be resolved using conventional SPA even for smaller complexes. Taken together, these findings broaden the potential of cryo-EM as a powerful tool for a variety of structure-based studies, particularly in drug discovery.

Though we could obtain a 3D reconstruction of iPKA$_c$ that allowed us to discern tertiary structural features, we were ultimately unable to achieve resolutions comparable to those of our ADH and mHb reconstructions. This was likely due to the lack of robust or accurate angular assignments of iPKA$_c$ particle images

aggregation in order to maintain a consistent ice thickness across the center of holes and provide sufficient signal for accurate CTF estimation, as observed previously[12].

**Cryo-EM data acquisition and image processing and refinement**. Microscope alignments were performed on a cross-grating calibration grid using methodologies previously described[12,14] with minor modifications. Briefly, after obtaining parallel illumination in diffraction mode at a diffraction length of D 850 mm, the length was increased to D 5.7 m and the intensity was adjusted to minimize the size of the caustic spot followed by minimization of the astigmatism of the diffraction lens. This process was iterated until no further improvements could be discerned visually. The resulting beam-intensity value was saved in Leginon for the exposure preset ($\times$73,000, 0.556 Å pixel$^{-1}$) and remained unchanged throughout data collection. The objective aperture was then centered, objective lens astigmatism was minimized, and coma-free alignment was performed using Leginon[13] as described previously[12,30]. Daily adjustments were made, if necessary, during data collection to maintain lens stigmation and to ensure the beam was centered properly. The hardware darks of the K2 Summit direct electron detector (Gatan) (DED) were updated approximately every 8–12 h.

All cryo-EM data were collected on a Thermo Fisher Scientific Talos Arctica transmission electron microscope (TEM) operating at 200 keV. All cryo-EM data were acquired using the Leginon[13] automated data collection program and all image pre-processing (e.g., frame alignment, CTF estimation, and initial particle picking) were performed in real-time using the Appion[31] image-processing pipeline. Movies were collected using a K2 Summit DED operating in counting mode (0.556 Å pixel$^{-1}$) at a nominal magnification of $\times$73,000 using a defocus range of −0.5 μm to −1.6 μm. Movies were collected over an 11 s exposure with an exposure rate of ~1.95 e$^{-}$ pixel$^{-1}$ s$^{-1}$, resulting in a total exposure of ~69 e$^{-}$ Å$^{-2}$ (1.57 e$^{-}$ Å$^{-2}$ frame$^{-1}$). Motion correction and dose-weighting were performed using the MotionCor2 frame alignment program[32] as part of the Appion pre-processing workflow. Frame alignment was performed on 5 × 5 tiled frames with a B-factor of 100 (ADH and metHb) or 250 (iPKA$_c$) applied. A running average of three frames was also used for frame alignment.

Unweighted summed images were used for CTF determination using CTFFIND4[33] and gCTF[34]. Specifically, CTFFIND4 within Appion was used for real-time CTF determination (512 box size, 40 Å minimum resolution, 3 Å max resolution, 0.10 amplitude contrast), wherein aligned micrographs with a CTF estimate confidence of fit below 90% were eliminated from further processing. Local CTF estimates using gCTF were then performed on the remaining micrographs using the standalone package (ADH) or a grid-based algorithm incorporated within Appion (metHb, PKA). For the latter, dummy coordinates were placed across the micrograph in rows and columns spaced 200 pixels apart and passed to gCTF for local CTF estimation using equiphase averaging (local box size of 512 pixels, 0.10 amplitude contrast, 50 Å minimum resolution, 4 Å max resolution, 1024-pixel field size). CTF values for a given particle were then determined using a cubic spline interpolation of the local CTF estimates within the grid.

For ADH, difference of Gaussian (DoG) picker[35] was used to automatically pick particles from the first 145 dose-weighted micrographs yielding a stack of 135,424 picks that were binned 4 × 4 (2.234 Å pixel$^{-1}$, 64-pixel box size) and subjected to reference-free 2D classification using RELION 2.1[19]. The best five classes that represented orthogonal views of ADH were then used for template-based particle picking using RELION. 1,232,543 picks were extracted from 1151 dose-weighted movies, binned 4 × 4 (2.234 Å pixel$^{-1}$, 64-pixel box size) and subjected to reference-free 2D classification using RELION using a 110 Å soft circular mask. The best 2D class averages that represented side or top/bottom views of ADH (i.e., the longest dimensions of ADH, see Fig. 1) were then isolated (639,430 particles). Those class averages that contained "end-on" or tilted views of ADH (e.g., the smallest dimension of ADH, see Fig. 1) were combined and subjected to another round of reference-free 2D classification using a 60 Å soft circular mask to focus on alignment of the smaller "end-on" views. The best 2D class averages were then selected (20,232 particles) and combined with the previously selected particles containing the longer side views for further processing.

A total of 659,662 particles corresponding to the best 2D class averages that displayed strong secondary-structural elements and multiple views of ADH were selected for homogenous ab inito model generation using cryoSPARC[26] to eliminate potential model bias. The generated model exhibited C2 symmetry and was low-pass filtered to 30 Å for use as an initial model for 3D auto-refinement in RELION. 659,662 binned 4 × 4 particles (2.234 Å pixel$^{-1}$, 64-pixel box size) were 3D auto-refined into a single class followed by subsequent re-centering and re-extraction binned 2 × 2 (1.117 Å pixel$^{-1}$, 256-pixel box size). Due to the close proximity of neighboring particles, any re-centered particle within a 30-pixel range of another was considered a duplicate and subsequently removed. These particles were then 3D auto-refined (C2 symmetry) into a single class using a scaled version of the binned 4 × 4 refined map. Upon convergence, the run was continued with a soft mask (5-pixel extension, 5-pixel soft cosine edge) around the entire molecule, followed by a no-alignment, 3D classification (6 classes, tau_fudge = 20) using the same soft mask. Particles comprising the best-resolved class was then subjected to 3D auto-refinement (C2 symmetry) using a soft mask. A subsequent no-alignment classification (2 classes, tau_fudge = 20) was performed and the class that possessed the best-resolved side-chain and backbone densities were re-centered and re-extracted unbinned (0.556 Å pixel$^{-1}$, 512-pixel box size). This final stack of 11,459 particles was 3D auto-refined

using a soft mask to ~2.92 Å (C2 symmetry) and ~3.45 Å (C1 symmetry) (gold-standard FSC at 0.143 cutoff[21]). Following per-particle defocus and beam tilt refinement using RELION 3.0, the final resolution of the reconstruction improved to ~2.72 Å (C2 symmetry) and ~3.14 Å (C1 symmetry), using phase randomization to account for the convolution effects of a solvent mask on the FSC between the two independently refined half maps[22,36].

For metHb, DoG picker[35] was used to automatically pick particles from the first 560 dose-weighted micrographs yielding a stack of 234,895 picks that were binned 4 × 4 (2.234 Å pixel$^{-1}$, 64-pixel box size) and subjected to reference-free 2D classification using RELION[19]. The best nine classes that represented orthogonal views of Hb were then used for template-based particle picking using RELION. 1,615,738 picks were extracted from 1673 dose-weighted movies, binned 4 × 4 (2.234 Å pixel$^{-1}$, 64-pixel box size) and subjected to reference-free 2D classification using RELION using a 80 Å soft circular mask. Class averages that displayed strong secondary-structural elements of metHb (513,632 particles) were combined and subjected to another round of reference-free 2D classification. A total of 160,169 particles corresponding to the best 2D class averages that displayed multiple views of tetrameric Hb were selected for homogenous ab inito model generation using cryoSPARC[26] to eliminate potential model bias. The generated volume was low-pass filtered to 20 Å and used as an initial model for 3D auto-refinement in RELION.

Binned 4 × 4 particles (2.234 Å pixel$^{-1}$, 64-pixel box size) were 3D auto-refined into a single class followed by subsequent re-centering and re-extraction binned 2 × 2 (1.117 Å pixel$^{-1}$, 128-pixel box size). Any re-centered particle within a 25-pixel range of another was considered a duplicate and subsequently removed. These particles were then 3D auto-refined (C2 symmetry) into a single class using a scaled version of the binned 4 × 4 refined map. Upon convergence, the run was continued with a soft mask (5-pixel extension, 5-pixel soft cosine edge) followed by three parallel no-alignment 3D classifications (four classes) with varying tau_fudge values (8, 12, or 20) using the same soft mask. Particles comprising the best-resolving classes across each classification were combined (35,809 particles) and duplicates were eliminated to maximize the subset of particles selected. The particles were then subjected to no-alignment classification (2 classes, tau_fudge = 20). Each class was then separately refined, re-centered, and re-extracted unbinned (0.556 Å pixel$^{-1}$, 256-pixel box size) and 3D auto-refined using a soft mask. Class 1 (24,308 particles) refined to a final estimated resolution of ~2.80 (C2 symmetry, ~3.17 Å resolution with C1 symmetry) and Class 2 (11,501 particles) refined to a final estimated resolution of ~3.25 Å resolution (C2 symmetry, ~3.48 Å resolution with C1 symmetry) according to gold-standard FSC[21] using phase randomization to account for the convolution effects of a solvent mask on the FSC between the two independently refined half maps[22,36]. Per-particle defocus and beam tilt refinement using RELION 3.0 did not yield improvements to the reconstructions.

Local resolution estimates for all reconstructions were calculated using the bloc_res function in BSOFT[23].

For PKA, DoG picker[35] was used to automatically pick particles from the untilted, aligned and dose-weighted micrographs yielding a stack of 554,170 picks that were binned 4 × 4 (2.234 Å pixel$^{-1}$, 64-pixel box size) and subjected to reference-free 2D classification using RELION[19]. A total of 314,001 particles corresponding to the best 2D class averages that displayed multiple views of iPKA$_c$ were selected for homogenous ab inito model generation using cryoSPARC[26] to eliminate potential model bias. The generated volume was low-pass filtered to 20 Å and used as an initial model for 3D auto-refinement (C1 symmetry) in RELION. Particles were then re-centered and re-extracted, Fourier binned 2 × 2 (1.117 Å/pixel, 128-pixel box) and subsequently 3D classified (tau_fudge = 4, E-step limit = 7 Å). Particles comprising the best-resolved class was 3D auto-refined to ~4.26 Å resolution, as estimated by gold-standard FSC. However, this EM density exhibited artifacts associated with preferred orientation (stretched features in directions orthogonal to the preferred view) and did not possess high-resolution features consistent with the Gold-standard FSC-estimated resolution of ~4.26 Å resolution. In an attempt to obtain the missing views of iPKA$_c$ and potentially improve the resolution of our reconstruction we collected a dataset at a tilt angle of 30° using methodologies previously described[27]. These aligned, dose-weighted micrographs were combined with the untilted data and forward projections of a 10 Å low-pass filtered EM density of iPKA$_c$ were used for template-based particle picking within RELION. Together, 1,762,088 particles were extracted, Fourier binned 4 × 4 (2.234 Å/pixel, 64-pixel box), and subjected to reference-free 2D classification (tau_fudge = 1, E-step limit = 7 Å) using RELION. It was apparent from the resulting 2D class averages that additional views of iPKA$_c$ were obtained from imaging a tilted specimen (Fig. 5). Particles comprising the "best" classes were 3D classified (tau_fudge = 1, E-step limit = 7 Å) and those classes that most resembled iPKA$_c$ were subjected to an additional round of 3D classification (tau_fudge = 1, E-step limit = 7 Å). The "best" resolved classes were then 3D auto-refined (C1 symmetry) to yield a ~4.57 Å resolution reconstruction, as estimated by gold-standard FSC. Although these particles refined to a similar resolution as the untilted data alone, the resulting reconstruction was more isotropic. Particles were re-centered and re-extracted, Fourier binned 2 × 2 (1.117 Å/pixel, 128-pixel box) and duplicate particle picks were eliminated. These particles were 3D auto-refined to ~6.24 Å resolution and subjected to no-alignment 3D classification (tau_fudge = 6). The "best" resolved classes were combined and 3D auto-refined to ~6.17 Å resolution. Refined particles with NrOfSignificantSamples values >25 were eliminated and the remaining particles were 3D auto-refined to ~4.34 Å resolution (as estimated by gold-standard FSC).

Although the additional views from the tilted dataset aided in lessening the artifacts resulting from preferred orientation, the final EM density does not possess molecular features consistent with the reported resolution of ~4.3 Å but rather resembles a ~6–7 Å resolution EM density. Strategies to further improve the quality of the reconstruction did not yield significant results. Such attempts included: (1) high-pass filtering the extracted particles to 100 or 120 Å prior to 3D refinements as has previously been implemented for high-resolution structure determination of small proteins[11]; (2) increasing the tau2_fudge value (e.g., 4, 6, 8, etc.) during refinement to place greater weight on the experimental data; (3) increasing the amplitude contrast of the inputted particle stack (e.g., 0.20 vs. 0.10 as used for the final structure); (4) filtering the particles iteratively by RELION metadata metrics (e.g., NrOfSignificantSamples, MaxValueProbDistribution, or Z-score) (5) using particle images with a box size just outside the bounds of the particle (e.g., 72 pixel box vs. 128-pixel box for binned 2 × 2 particles); or (6) various combinations of the above mentioned.

**Model building and refinement**. For each of the final reconstructions, an initial model that had been stripped of all ligands, waters, and alternative conformations, had all occupancies set to zero, had a single B-factor value was set for all atoms, and had all Ramachandran, rotameric, and geometric outliers corrected, was subjected to a multi-model pipeline using methodologies similar to those previously described[37]. Briefly, PDB ID: 2JHF [https://doi.org/10.2210/pdb2JHF/pdb] was used as the starting model for ADH, chains A-D from PDB ID: 4N7P [https://doi.org/10.2210/pdb4N7P/pdb] was used for metHb class 1, and chains E-H from PDB ID: 4N7N [https://doi.org/10.2210/pdb4N7N/pdb] were used for metHb class 2. These initial models were then refined into the EM density using Rosetta[38] while enforcing the same symmetry that was applied during reconstruction and adjusting the Rosetta weighting and scoring functions according to the FSC-estimated map resolution. Each of the 200 Rosetta-refined models were then ranked based on the number of Ramachandran outliers, geometry violations, Rosetta aggregate score, and MolProbity clashscore[39]. The 10 structures that scored the best across all categories were selected for further real-space refinement using the Phenix refinement package[40] after incorporating cofactors (e.g., NADH for ADH and heme for metHb) and active-site and structural Zn ions (ADH). Model–model agreement statistics are based on the per-residue Cα RMSD[37].

**Reporting summary**. Further information on experimental design is available in the Nature Research Reporting Summary linked to this article.

## Data availability

Data supporting the findings of this manuscript are available from the corresponding author upon reasonable request. A reporting summary for this Article is available as a Supplementary Information file. The atomic coordinates for the ADH and metHb (class 1 and class 2) structures have been deposited in the Protein Data Bank (PDB) under accession codes 6NBB [https://doi.org/10.2210/pdb6NBB/pdb], 6NBC [https://doi.org/10.2210/pdb6NBC/pdb], and 6NBD [https://doi.org/10.2210/pdb6NBD/pdb], respectively. The corresponding EM density maps (final unsharpened and sharpened maps, half maps, and masks) have been deposited to the Electron Microscopy Data Bank under accessions EMD-0406, EMD-0407 and EMD-0408, respectively, and EMD-0409 for iPKAc. Uncorrected movie frames and associated gain correction images for ADH, metHb, and iPKAc datasets are available on the EMPIAR under accessions 10249, 10250, and 10252, respectively.

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

## Acknowledgements

We thank J.C. Ducom at The Scripps Research Institute (TSRI) High Performance Computing facility for computational support, B. Anderson at the TSRI electron microscopy facility for microscope support, and M. Vos for advice and discussion regarding TEM alignments. We thank P. Aoto of the S. Taylor laboratory (University of California, San Diego) for kindly providing iPKA$_c$ for this study. G.C.L. is supported as a Searle Scholar and as a Pew Scholar, by a young investigator award from Amgen, and by the US National Institutes of Health (NIH) grant DP2EB020402. Computational analyses of EM data were performed using shared instrumentation at TSRI funded by NIH S10OD021634. M.A.H. is supported by a Helen Hay Whitney Foundation postdoctoral fellowship. M.W is supported by a National Science Foundation Graduate Student Research Fellowship.

## Author contributions

M.A.H. and M.W. performed all cryo-EM experiments and analyses under the supervision of G.C.L., M.A.H., M.W., and G.C.L. contributed to the experimental design and manuscript preparation.

## Additional information

**Competing interests:** The authors declare no competing interests.

