## [Peer Review File · Nature Communications]

Reviewers' Comments:

Reviewer #1:

Remarks to the Author:

In the manuscript entitled "High-resolution structure determination of sub-100 kilodalton complexes using conventional cryo-EM" the authors Herzik, Wuy and Lander built upon their publication from 2017. Here, they had originally shown that high-resolution structure determination is in fact possible using microscopes that had previously been considered "screening machines".

In the current manuscript the authors now demonstrate that the same approach is possible even for very small particles, which were previously considered not to be amenable to cryo-EM. The paper is written very clearly and logical and the data is of very high quality. However, the results that are presented are somewhat not too surprising anymore. First of all, other groups have now also shown that it is possible to obtain high-resolution from small samples (see Fan et al. 52kDa, a paper that is correctly acknowledged by the authors) and secondly, considering the previous manuscript from 2017 this result is rather incremental.

In my view the methods section is the real highlight of this manuscript and I would wish that every manuscript could be as detailed as this. I am certain that the protocols described there will be a great resource for many labs. Therefore, this demonstration will be of high interest to the cryo-EM community and deserves to be published.

I have a few comments that should be considered:

1. Even though more expensive TEMs are the standard for high-resolution data acquisition the EM that was utilized in this study is still far from being "conventional". A conventional TEM is a 120keV machine with a Lab6 electron source, a side-entry holder and a simple digital camera. The machine that was used by the authors will in many cases still be the most expensive equipment in the biology department, or even at the entire university. Therefore, I think the title is not well chosen and misleading.
2. In the general introduction the authors state that 3 complexes have been solved to better than 4 Å resolution and later in that paragraph they set a 5 Å barrier. I would suggest to use the same level for comparison.
3. The authors state that 99% of all structures better than 5 Å are amassing > 200kDa. This is simply wrong. There are approximately 2000 structures with higher resolution than 5 Å in the database. Therefore, only 40 are allowed to weigh less than 200 kDa, if this claim would be true. I went through the database and was readily able to identify significantly over 50 structures that would fit this criterion. If the authors desire to make such a strong claim they have to be accurate!
4. To obtain their structure of ADH the authors went through iterations of 2D classification. After an initial round, different views are separated to precisely adjust the circular mask according to the diameter of a top and end on view, respectively. It is not clear to me whether this was necessary to obtain a high resolution 3D density. Was this masking carried over to the 3D refinement and classification or was it simply to obtain better (purer) classes?
5. I agree, with the authors that sample thickness is a major factor for their success. The important question is here: What is "thin". Is it just thick enough to incorporate the proteins of interest or is it still significantly thicker. The latter seems to be more likely as the authors also report strong orientationally bias which indicates interactions at the air-water interface. As the Leginon system was utilized for imaging it should be possible to elegantly address this question and to measure the sample thickness and the orientation/position of the particles.
6. Space missing in figure caption 2 " ... BSOFT23)_and subunit ..."

Reviewer #2:

Remarks to the Author:

In this work, the authors push to their limits the capabilities of conventional defocus phase contrast cryo-electron microscopy. They achieved several remarkable cryo-EM structures at the frontier of what is currently possible in terms of molecular weight and ligand size. The portfolio of results is a perfect demonstration of the current capabilities of cryo-EM and will become a guideline and a benchmark to researchers considering the technique, newcomers to the field and skilled artisans alike. It will increase the confidence of investigators in the conventional cryo-EM approach and prevent the unnecessary use of advanced techniques, such as phase plates, in cases where the possible benefits are outweighed by the increased chance of failure. The paper sets a new standard in cryo-EM and structural biology and I strongly recommend its publication in Nature Communications, with a few minor questions and recommendations.

1. I would recommend including B-factor plots for the ADH and Hb datasets calculated with the random particle subsets method, aka Rosenthal & Henderson plots. Relion 3 includes a script "bfactor_plot.py" that automates this task. The B-factors calculated in this manner represent much better the overall performance than resolution numbers or the sharpening B-factors.
2. The angular accuracies (rlnAccuracyRotations) reported in the run_model.star file after 3D refinement are also a very good indicator of the "alignability" of particles. Probably even more so than the number of significant samples histograms in Supp. Fig. 5. If possible, please include the angular accuracy values in Table 1.
3. There seems to be a trend with very small molecules that very few of the picked particles are retained in the final 3D refinement. This is the case with the ADH and Hb datasets, as well as the streptavidin data in ref. 11. Do the authors have a hypothesis about this? Is it due to gross overpicking, more damaged particles at the air-water interface or classification for particles with the most "suitable" noise? Are the retained particles equally distributed across all micrographs or are they from a few very good images?
4. During the Hb data processing, what was the motivation behind 3 rounds of no-alignment 3D classification with increasing tau, instead of a single round with tau=20?
5. Did the authors try the new CTF refinement and Bayesian polishing routines in Relion 3? If yes, how did they perform with such small molecules?
6. The supplementary .docx file provided in addition to the main PDF file seems to contain different versions of the supp. figures, e.g. Supp. Fig. 3 has different panel notation and does not contain the putative water molecules panel (h). Please make sure to submit the correct version during the final submission.
7. Page 4, right column, last paragraph: There is no information about the Hb heterodimers in Supp. Fig. 3.
8. Page 5, left column, last paragraph: Supp. Fig. 3c -> 4c.
9. Methods, page 7, left column, last paragraph: "... freshly plasm cleaned ..." -> "... freshly plasma cleaned ..."

We thank the reviewers for their speedy assessment of our manuscript and providing their feedback and helpful comments. Their constructive criticisms and questions are pasted below in italics, and our responses follow.

Reviewer 1

1. Even though more expensive TEMs are the standard for high-resolution data acquisition the EM that was utilized in this study is still far from being “conventional”. A conventional TEM is a 120keV machine with a Lab6 electron source, a side-entry holder and a simple digital camera. The machine that was used by the authors will in many cases still be the most expensive equipment in the biology department, or even at the entire university. Therefore, I think the title is not well chosen and misleading.

We thank the reviewer for pointing out this criticism, and we see their side, as there was discussion about this point among the authors prior to submission. We are not referring to the microscope itself as conventional (which would have warranted the title “...complexes using a conventional cryo-EM”), but rather the defocus-based approach for imaging, which is the conventional method for data acquisition regardless of the type of electron microscope used.

2. In the general introduction the authors state that 3 complexes have been solved to better than 4 Å resolution and later in that paragraph they set a 5 Å barrier. I would suggest to use the same level for comparison.

We used these different resolution limits to emphasize different points. Ab initio modeling is possible at better than 4 Å, and this is the minimal resolution target for atomic model generation, which is why we used this cutoff to emphasize how few structures of this size are in the database. The more generous 5 Å resolution cutoff was used to demonstrate the dearth of small structures that are resolved at even moderate resolution.

3. The authors state that 99% of all structures better than 5 Å are amassing > 200kDa. This is simply wrong. There are approximately 2000 structures with higher resolution than 5 Å in the database. Therefore, only 40 are allowed to weigh less than 200 kDa, if this claim would be true. I went through the database and was readily able to identify significantly over 50 structures that would fit this criterion. If the authors desire to make such a strong claim they have to be accurate!

We stand by the accuracy of our stated statistics, as we carefully examined the structures reported to be both smaller than 200 kDa and at better than 5 Å resolution in the EMDB. Using default search parameters will include structures determined by micro-electron diffraction, which yields numerous high-resolution structures of peptides or very small proteins. Further, there are a number of structures in the EMDB with incorrectly assigned molecular weights, or the weight of the asymmetric unit (such as a protomer within a large icosahedral virus) has been assigned as the weight for the entry (e.g. EMDB-3073, EMDB-3074). We only considered entries where the entire deposited macromolecule weighed less than 200 kDa and was determined by single particle cryo-EM.

4. To obtain their structure of ADH the authors went through iterations of 2D classification. After an initial round, different views are separated to precisely adjust the circular mask according the diameter of a top and end on view, respectively. It is not clear to me whether this was necessary to obtain a high resolution 3D density. Was this masking carried over to the 3D refinement and classification or was it simply to obtain better (purer) classes?

This has been clarified in the text.

5. I agree, with the authors that sample thickness is a major factor for their success. The important question is here: What is “thin”. Is it just thick enough to incorporate the proteins of interest or is it still significantly thicker. The latter seems to be more likely as the authors also report strong orientationally bias which indicates interactions at the air-water interface. As the Leginon system was utilized for imaging it should be possible to elegantly address this question and to measure the sample thickness and the orientation/position of the particles.

We think the reviewer mistakenly wrote “latter” as opposed to “former”, as thin ice that is barely thick enough to incorporate the protein sample is more likely to give rise to strong orientation bias as opposed to thicker specimens. Because we don’t have an energy filter on our Arctica, we attempted to estimate the ice thickness of the acquired images based on the relative image intensities using the formula described in Rice et al. JSB 2018 (thickness = $\Lambda \ln I_0/I$), assuming Λ (mean free path) = 300 nm (as specified in the paper for 200 keV, 70 μ M objective aperture). While this may work well for the thicker ice specimens that are used as examples in the Rice paper, it did not work well for our data, as the results of our calculations suggested that the ice thickness was 0. To give a sense of the ice thickness on our grids, we have included lower magnification images in the supplement.

6. Space missing in figure caption 2 “... BSOFT23)_and subunit ...”

This has been corrected.

Reviewer 2

1. I would recommend including B-factor plots for the ADH and Hb datasets calculated with the random particle subsets method, aka Rosenthal & Henderson plots. Relion 3 includes a script “bfactor_plot.py” that automates this task. The B-factors calculated in this manner represent much better the overall performance than resolution numbers or the sharpening B-factors.

We agree with the reviewer on this point, and now include the b-factor plots for the ADH and Hb datasets in the revised manuscript.

2. The angular accuracies (*rlnAccuracyRotations*) reported in the *run_model.star* file after 3D refinement are also a very good indicator of the “alignability” of particles. Probably even more so than the number of significant samples histograms in Supp. Fig. 5. If possible, please include the angular accuracy values in Table 1.

These values are now added to Table S1.

3. There seems to be a trend with very small molecules that very few of the picked particles are retained in the final 3D refinement. This is the case with the ADH and Hb datasets, as well as the streptavidin data in ref. 11. Do the authors have a hypothesis about this? Is it due to gross overpicking, more damaged particles at the air-water interface or classification for particles with the most “suitable” noise? Are the retained particles equally distributed across all micrographs or are they from a few very good images?

We have ideas about this interesting topic but deemed them too speculative to include in the main text. We have included a supplemental note containing our hypotheses about this.

4. During the Hb data processing, what was the motivation behind 3 rounds of no-alignment 3D classification with increasing tau, instead of a single round with tau=20?

Due to the relatively small subsets of particles corresponding to the best-resolved classes from no-alignment 3D classification, which also varied depending on the tau_fudge values used, we sought to maximize the number of particles out of 3D classification by selecting for the unique particles across three rounds of classification performed with increasing tau. This method yielded the best reconstructions over those obtained from subsets selected from any single round of 3D classification.

5. Did the authors try the new CTF refinement and Bayesian polishing routines in Relion 3? If yes, how did they perform with such small molecules?

At the reviewer’s suggestion we performed the per-particle CTF refinement routine implemented within RELION 3.0 (defocus UV and whole micrograph astigmatism), which yielded the following results:

Alcohol Dehydrogenase: improvement of 0.2 Å (resolution = ~2.7 Å), estimated beam tilt: -0.027 mrad (x), 0.22 mrad (y).

metHb: no improvement (resolution = ~2.8 Å), estimated beam tilt: 0.032 mrad (x), 0.027 mrad (y).

We have included these results in the Methods section and have updated the figures containing alcohol dehydrogenase accordingly. We did not perform the Bayesian polishing routine implemented within RELION 3.0 as that would require a substantial time to retrieve our raw data from archive and process to completion, and delay our resubmission by at least 1 month. Although the results may potentially benefit the field, we don't anticipate a substantial increase in resolution and the results will not change our manuscript's main conclusion that high-resolution structures of <100 kDa proteins can be determined without phase plates or other imaging accessories. We have deposited the raw data to the Electron Microscopy Public Image Archive Repository (EMPIAR) server and we hope that other groups will re-process our data using the most-recent algorithms, including the Bayesian polishing routine implemented within RELION, and report their results to the community.

6. The supplementary .docx file provided in addition to the main PDF file seems to contain different versions of the supp. figures, e.g. Supp. Fig. 3 has different panel notation and does not contain the putative water molecules panel (h). Please make sure to submit the correct version during the final submission.

This has been corrected.

7. Page 4, right column, last paragraph: There is no information about the Hb heterodimers in Supp. Fig. 3.

8. Page 5, left column, last paragraph: Supp. Fig. 3c -> 4c.

This has been corrected.

9. Methods, page 7, left column, last paragraph: "... freshly plasm cleaned ..." -> "... freshly plasma cleaned ..."

This has been corrected.

Reviewer 3

1. The authors made cryo-specimens with very thin layer of ice to enhance the signal to noise ratio. A major concern would be the artifact to the protein structures caused by the air-water interface. It appeared that only very small fraction of single particle images (<2% of the total picked particles) contributed to the final high resolution structure reconstruction. Could the authors provide more information about the protein molecule's distribution in the ice for instance using electron tomography?

We agree that we should have collected tomograms to assess ice thickness during data acquisition, and will do so in the future. The grids we used for collection were not saved, and thus the only means of retroactively assessing ice thickness is to estimate the ice thickness using methodologies outlined in Rice *et al.* JSB 2018. However, the approach did not yield consistent or meaningful values (as detailed above for Reviewer #1), suggesting that the ice for the ADH and metHb datasets is either barely thicker than the diameter of the particles, or that an energy filter is required to accurately measure such thin layers of ice. We have also added a supplemental note discussing the issue of ice thickness and particle quality.

2. Related to the above comment, the ice thickness of the usable area for high resolution data collection would be very useful to guide other researchers' work. The authors may provide some EM images of the grid atlas, the typical holes under low and intermediate magnification with the usable areas labeled.

For each of the samples detailed in this manuscript, we have provided representative medium-mag (i.e. 1200x magnification) images. High-mag (i.e. exposure, 73000x magnification) are also depicted in the processing

figures in the supplement. We hope these images inform the audience on the thickness of ice we targeted for data collection.

3. The authors described that they used a 3D reconstruction of the iPKAc low-pass filtered to 10 Angstrom to perform particle picking from the tilted micrographs of iPKAc. This leads to a concern of model bias during the particle picking from the micrographs with very low signal-to-noise ratio. What if the authors use a 3D reconstruction low-pass filtered to even lower resolution such as 30 Angstrom as reference for the particle picking? Can they just use the reference-free 2D class averages calculated from the untilted micrographs for particle picking in the tilted micrographs?

The authors agree with the reviewer that template-based particle-picking is prone to model bias and false positives. To potentially ameliorate these effects, we employed numerous rounds of template-based particle picking using templates generated with various levels of low-pass filtering and carefully evaluated micrograph coverage and over-picking before deciding on final parameters for the entire dataset. We concluded that a 10 Å low-pass filtered initial model provided the most coverage without substantial over-picking. More aggressive low-pass filters (e.g. > 20 Å) did not provide adequate coverage of the micrographs, presumably, due to the lack of features present in the templates. The authors also attempted particle picking using 2Ds from the untilted data but these templates resulted in too few picks per micrograph, presumably, due to non-overlapping views.

4. The authors used very high concentration of protein solution to make vitrified grids. Could the authors rationalize the reason for the high concentration?

As mentioned in our previous manuscript (Herzik, Wu, & Lander Nat Methods 2017) as well as in Noble *et al.* eLife 2018, the benefits of imaging smaller macromolecules at higher particle densities include increased signal for estimation of CTF parameters as well as improved signal-to-noise for motion correction during frame alignment. Anecdotally, the authors have concluded that the imaging of smaller (<200 kDa) macromolecules in thin ice at lower protein concentrations results in more rapid desorption of the ice, causing the ice across the hole to “pop” during image acquisition. We speculate that this results from uneven expansion of the thin ice, causing the microscopic tears that lead to destabilization and “popping” of the fragile ice layer.

5. The size of objective aperture for data collection should be provided.

We have included the size of the objective aperture in Table S1 of the revised manuscript (100 μm)

6. For MotionCor2 frame alignment, the authors used a B-factor of 100 for ADH and Hb but 250 for PKA. Is there a relationship of B-factor to the size of the molecule?

Although we have not extensively examined the contribution of B-factors used during frame alignment for a wide range of particles of various sizes and symmetries, B-factor scouting during motion correction for the ADH, metHb, and PKA datasets resulted in optimal B-factors of 100, 100, and 250 Å², respectively, with PKA additionally benefiting from a running average window of 3 frames. The authors feel that these values should be optimized for each dataset and particles of lower molecular weights will most likely benefit from the application of higher B-factors and multi-frame averaging during frame alignment.

7. There are typos throughout the manuscript that need to be proofread carefully.

We apologize for these typos and have gone through the text thoroughly to correct these errors. We anticipate that the editorial staff at Nature Communications will catch remaining typos, if any, that we missed.

Reviewers' Comments:

Reviewer #1:

Remarks to the Author:

I do not have any comments that should delay publication. Therefore, I would accept this manuscript without further modification.

Reviewer #2:

Remarks to the Author:

The authors answered all questions and took into account all suggestions.

Radostin Danev

Reviewer #3:

Remarks to the Author:

The manuscript entitled "High-resolution structure determination of sub-100 kilodalton complexes using conventional cryo-EM" reports the high-resolution reconstruction of three protein complexes smaller than 100 kDa using single particle cryo-EM on a 200 kV TEM. The authors optimized the microscopy illumination conditions, data collection and image processing strategies and proved the capability of high-resolution structure determination of sub-100 kDa protein complexes on a conventional cryo-EM imaging condition. This work will be of great interest to the structural biology field in general and pharmaceutical industry. The manuscript was well-written with clear logics and demonstration. The revised manuscript has addressed most of the reviewers' comments adequately.

A few minor points for the authors to correct.

1. The first sentence in the Methods:

"Sampling handling and protein purification" should be "Sample handling and protein purification".

2. Supplementary Figure 3, the figure legend should be revised to reflect the figure panels correctly.

3. Supplementary Figure 4d, the number of particles after the first two rounds of 3D classification is 308,534, more than the summation of particles from the selected three classes. This should be clarified.

4. Because the reported resolution by the FSC does not reflect the true resolution of the Protein kinase A map as the authors concluded, the local resolution of the map or the FSC between the map and the docked atomic model should be included to reflect this fact.

5. In the Supplementary Note, the air-water interface interaction should also be included as one of the factors that cause particles to be excluded from the final high-resolution reconstruction.

Responses to reviewer 3 are in red

Reviewer #3 (Remarks to the Author):

The manuscript entitled “High-resolution structure determination of sub-100 kilodalton complexes using conventional cryo-EM” reports the high-resolution reconstruction of three protein complexes smaller than 100 kDa using single particle cryo-EM on a 200 kV TEM. The authors optimized the microscopy illumination conditions, data collection and image processing strategies and proved the capability of high-resolution structure determination of sub-100 kDa protein complexes on a conventional cryo-EM imaging condition. This work will be of great interest to the structural biology field in general and pharmaceutical industry. The manuscript was well-written with clear logics and demonstration. The revised manuscript has addressed most of the reviewers’ comments adequately.

A few minor points for the authors to correct.

1. The first sentence in the Methods:

“Sampling handling and protein purification” should be “Sample handling and protein purification”.

This has been corrected

2. Supplementary Figure 3, the figure legend should be revised to reflect the figure panels correctly.

This has been corrected

3. Supplementary Figure 4d, the number of particles after the first two rounds of 3D classification is 308,534, more than the summation of particles from the selected three classes. This should be clarified.

We thank the eagle-eyed reviewer 3 for noticing this, the figure has been updated with the correct particle numbers

4. Because the reported resolution by the FSC does not reflect the true resolution of the Protein kinase A map as the authors concluded, the local resolution of the map or the FSC between the map and the docked atomic model should be included to reflect this fact.

We now include the FSC between the map and the docked atomic model in Supplementary Figure 4

5. In the Supplementary Note, the air-water interface interaction should also be included as one of the factors that cause particles to be excluded from the final high-resolution reconstruction.

We included the air-water interaction in paragraph 4 of the Supplementary note:

“...destabilizing or denaturing interactions at the air-water interface,”